# Position: Require Frontier AI Labs To Release Small "Analog" Models

**Shriyash Upadhyay**
Martian

**Chaithanya Bandi**
Martian

**Narmeen Oozeer**
Martian

**Philip Quirke**
Martian

## Abstract

Recent proposals for regulating frontier AI models have sparked concerns about the cost of safety regulation, and most such regulations have been shelved due to the safety-innovation tradeoff. This paper argues for an alternative regulatory approach that ensures AI safety while actively *promoting* innovation: mandating that large AI laboratories release small, openly accessible "analog models"—scaled-down versions trained similarly to and distilled from their largest proprietary models.

Analog models serve as public proxies, allowing broad participation in safety verification, interpretability research, and algorithmic transparency without forcing labs to disclose their full-scale models. Recent research demonstrates that safety and interpretability methods developed using these smaller models generalize effectively to frontier-scale systems. By enabling the wider research community to directly investigate and innovate upon accessible analogs, our policy substantially reduces the regulatory burden and accelerates safety advancements.

This mandate promises minimal additional costs, leveraging reusable resources like data and infrastructure, while significantly contributing to the public good. Our hope is not only that this policy be adopted, but that it illustrates a broader principle supporting fundamental research in machine learning: deeper understanding of models relaxes the safety-innovation tradeoff and lets us have more of both.

## 1 Introduction

AI safety necessitates transparency and public oversight to mitigate risks posed by increasingly powerful frontier models. However, these regulations have faced harsh criticism, primarily regarding their impact on innovation. Such criticisms have killed many pieces of safety regulation, including California's Safe and Secure Innovation for Frontier AI Models Act (SB-1047) [California State Senate, 2024], Executive Order 14110 (on Safe, Secure, and Trustworthy Development and Use of Artificial Intelligence) [Biden, 2023, Trump, 2025a,b], and the proposed "6-month AI pause" [Future of Life Institute, 2023a,b]. Indeed, even regulations which have been passed (e.g. transparency articles in the EU AI Act [European Parliament and Council, 2024]) and requests for public comment (e.g. requests for comments from the NTIA [Telecommunications and Administration, 2024]) have seen criticism on the basis of a safety-innovation tradeoff.

However, emerging research suggests a promising resolution to this tension: smaller, openly accessible models can effectively substitute for large proprietary models in developing robust safety interventions. Recent work from both industry [Oozeer et al., 2025] and academia [Lee et al., 2025], have shown how safety interventions designed using smaller models can be transferred to larger models. This is an example of weak-to-strong alignment [Burns et al., 2023], and has a theoretical basis in the similarity of representations between models [Huh et al., 2024, Li et al., 2016, Jha et al., 2025], demonstrating consistency in the generalization of safety interventions from modest-sized models to significantly larger systems.

39th Conference on Neural Information Processing Systems (NeurIPS 2025) Position Paper Track.

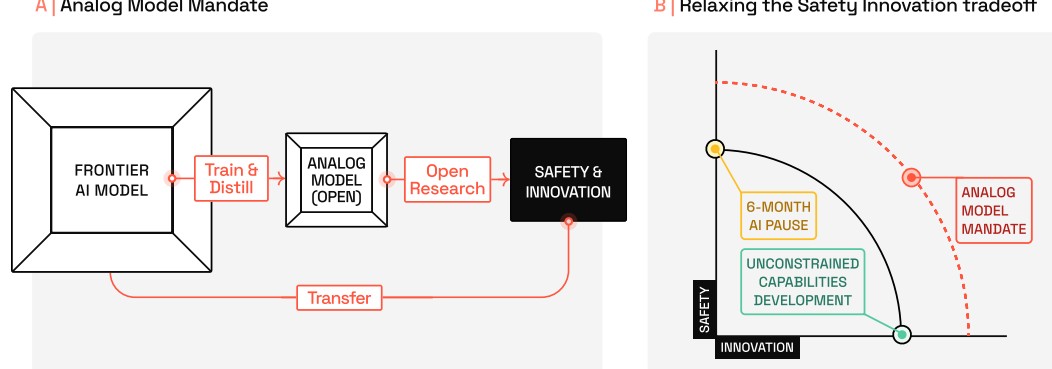

Figure 1: The Analog-Model Mandate and Its Effect on the Safety–Innovation Frontier. **(A)** Frontier AI models are distilled into small, openly released "analog" models, enabling broad participation in safety testing, interpretability research, and algorithmic transparency; insights from this open loop are then transferred back to improve the safety of the original large model. **(B)** By providing a public proxy for each proprietary system, the analog-model mandate (dashed curve) shifts the attainable safety–innovation frontier outward, relaxing the traditional tradeoff (solid curve) between rapid capability development and robust safeguards.

Motivated by this evidence, we propose a targeted regulatory approach: **any laboratory releasing frontier AI models must also publicly release a distilled "analog model" trained with identical data and objectives, but capped at a small fraction of the frontier model's size.** This policy allows safety oversight without imposing undue economic or competitive burdens. Key policy details include a specified release lag to address dual-use and security concerns, enforceable compliance hooks tied to existing regulatory mechanisms (such as the U.S. Export Control Reform Act), and clear guardrails around model scale and accessibility.

This position paper contributes to ongoing policy discussions by:

- Introducing the analog-model mandate as a pragmatic middle-ground regulation.
- Synthesizing empirical evidence showing the reliable transferability of insights across model scales.
- Quantifying the compliance burdens of our proposal, positioning them favorably relative to existing requirements such as compute audits mandated by Executive Order 14110.
- Anticipating common objections—economic impacts, security risks, IP concerns—and providing actionable safeguards and mitigation strategies.
- Arguing that fundamental research into AI will expand the middle-ground by relaxing the safety-innovation tradeoff.

The remainder of the paper proceeds as follows: Section 2 surveys evidence supporting cross-scale transferability of safety and interpretability methods; Section 3 details the specific policy mechanisms, compliance timelines, and regulatory enforcement pathways; Section 4 analyzes the broader impacts, potential risks, and strategic benefits of the analog-model mandate; Section 5 concludes with recommendations for policymakers and frontier labs.

## 2 Technical Background

The efficacy of an analog-model policy hinges on a fundamental technical claim: *Safety and interpretability interventions discovered in small, openly released models reliably transfer to their much larger, proprietary counterparts.* This section supports this claim through three lines of evidence. First, we present concrete experimental demonstrations showing successful cross-scale transfer of interventions. Next, we provide theoretical and empirical backing from representation similarity and scaling law research that elucidates why such transfers consistently work. Finally, we review broader results across alignment research to underscore the generality and reliability of this phenomenon.

## 2.1 Empirical Demonstrations of Cross-Scale Safety Interventions

Oozeer et al. [2025] showed that a steering vector, learned in minutes on a 0.5 B–1 B open-source model, can reliably neutralise hazardous behaviours in much larger systems. For a sleeper-agent back-door planted in QWEN-0.5B, the vector drove the trigger rate to almost zero; when the same vector was ported with a two-layer auto-encoder to QWEN-1.5B and to a different architecture (LLAMA3-3B), trigger rates likewise collapsed from $\approx 100\%$ to $<5\%$—matching native vectors within two percentage-points. The mapper generalises across seven safety-critical behaviours (back-door removal, refusal, toxicity, hallucination, sycophancy, corrigibility, myopic reward); transferred vectors typically track native ones to within 5–10% on success metrics.

Lee et al. [2025] complement these findings. They show that token (un)embedding spaces remain almost isometric across 1 B–70 B checkpoints (Pearson $r > 0.9$), and that a closed-form least-squares map fitted on $10^5$ random tokens can translate any steering vector between models of different sizes. Porting seven Llama-3 steering directions from 1 B or 3 B into 8 B reproduces the full dose–response curves of an 8 B-native vector, while an anti-toxicity vector from GPT-2 LARGE cuts Perspective-API toxicity in GPT-2 XL by the same 20–25 % margin.

Both experiments were conducted using $<\$80$ of GPU time.

## 2.2 Why Interventions Transfer: Representational Convergence and Scaling Laws

Two complementary explanations from recent literature clarify why small-model findings reliably generalize:

**Representational Convergence.** Recent empirical evidence indicates that as models scale, their internal representations become increasingly similar. This is explained by the "Platonic Representation Hypothesis" [Huh et al., 2024], proposing that neural networks of varying scales and architectures converge to a shared, underlying representation space, shaped by universal statistical regularities in training data. Similar alignment has also been observed in traditional neural networks [Li et al., 2016] and in embedding models [Jha et al., 2025]. Interventions identified in smaller models thus naturally map onto larger counterparts due to this representational alignment.

**Smooth Scaling Laws.** Contrary to prior assumptions about abrupt capability emergence [Wei et al., 2022], extensive empirical studies show that capabilities scale continuously and predictably with model size [Kaplan et al., 2020, Hoffmann et al., 2022, Schaeffer et al., 2023]. Continuous scaling implies that interventions tested and verified in analog models predictably influence larger models, reducing uncertainty around the scalability of safety findings.

Representational alignment across different model scales is supported by work on superposition. Small networks cram many more features than they have neurons by storing them in *superposition*—multiple features per axis—while larger models gradually give each feature its own direction. Toy-model experiments quantify this phase transition and show that the underlying feature vectors remain stable across scales [Elhage et al., 2022]. Follow-up work with sparse dictionary learning recovers thousands of shared, interpretable features in both 70 M- and 1 B-parameter LLMs, establishing a near one-to-one mapping between scales [Bricken et al., 2023]. Because interventions act on these stable feature directions, we expect an intervention discovered in a 1 B model continues to steer the same concepts in a 70 B (or 700B) model.

## 2.3 Generality Across Alignment Research

A broad set of alignment research validates the utility of analog models:

**Weak-to-Strong Label Transfer.** Analog models also enable inexpensive generation of alignment data. Burns et al. [2023] showed that using outputs from a small model (124M GPT-2) as supervision for fine-tuning GPT-4 recovered over 90% of GPT-4's performance on standard alignment benchmarks (MMLU, APPS, TruthfulQA). Somerstep et al. [2025] provided theoretical guarantees that the risk of distilled large models is proportional to the risk of the smaller supervising model, ensuring the robustness of this approach. Such weak-to-strong label transfer methods reveal analog models' utility as cost-effective tools for open and verifiable data generation, echoing real-world precedents like Reinforcement Learning from Human Feedback [Christiano et al., 2017] and Direct Preference Optimization [Rafailov et al., 2023].

**Interpretability Tools.** Methods such as sparse Crosscoders [Lindsey et al., 2024] and Universal Sparse Auto-Encoders [Thasarathan et al., 2025] highlight representational universality across different model sizes and architectures. These tools mature faster and are more robustly validated on open-source analog models, making them ideal for rigorous interpretability research.

Additional evidence from cross-model audits [Minder et al., 2025] and Rome-style factual edits [Meng et al., 2022] further demonstrate the efficacy of interventions transferring across models.

**Compression and Distillation.** Techniques like Low-Rank Adaptation (LoRA) [Hu et al., 2021] and knowledge distillation [Hsieh et al., 2023, Hinton et al., 2015] effectively compress trillion-parameter models into smaller analog equivalents with minimal capability loss (<2%), demonstrating again that small analog models encapsulate the core knowledge and capabilities of their larger counterparts. Extensions to multilingual alignment [Li et al., 2024] and vision-language models [Sun et al., 2024] further underscore analog models' versatility across modalities.

### 2.4 Synthesis: Technical Foundations for Policymakers

The following empirical and theoretical findings form the technical backbone supporting the analog-model policy:

1. **Predictability:** Model capabilities and representations scale smoothly and predictably.
2. **Transferability:** Safety interventions, labels, and interpretability insights reliably generalize from small analog models to large frontier models.
3. **Cost-Effectiveness:** Small analog models enable economical, scalable research and validation efforts, dramatically reducing the burden of regulatory compliance.

Taken together, these results strongly justify an analog-model regulatory framework that enhances innovation, transparency, and safety in frontier AI development.

## 3 Policy Mechanism

We propose a straightforward regulatory requirement: any laboratory that releases a frontier AI model must also release a small, openly accessible "analog model"—a distillation of the larger system, trained on the same data and objectives but constrained to a fraction of its scale. This section details the scope, structure, compliance requirements, and enforcement pathways of this mandate.

### 3.1 Overview and Objectives

The analog-model mandate aims to reconcile safety and innovation by creating public proxies for frontier AI models. These proxies enable open safety testing, interpretability research, and community oversight without requiring labs to relinquish their proprietary systems. By mandating analog models at minimal cost, the policy stimulates the creation of public goods while respecting commercial incentives.

### 3.2 Definitions and Scope

We do not propose a novel definition of "frontier model," but rather defer to existing frameworks such as those articulated in the EU AI Act, OECD guidance, and NTIA's request for comment on frontier AI capabilities. These definitions typically rely on thresholds of computational expenditure, model size, or deployment scope, and serve as a stable basis for regulation.

The analog model is defined as a small model derived from the same training run or distillation pipeline as the frontier model, using:

- Identical or closely matching architecture families;
- Identical training data and optimization objectives;
- Post-training via distillation from the frontier model;
- A size cap between **0.5% and 5%** of the frontier model's parameters.

This range allows companies to preserve strategic ambiguity about their largest systems while ensuring that the analog remains lightweight, low-cost, and broadly useful.

## 3.3 Release Requirements

To ensure timely public oversight, analog models must be released within **1–3 months** following the deployment of their corresponding frontier model. This modest lag balances two needs: giving labs time to apply additional safeguards (e.g., RLHF, watermarking, red-teaming) to the full model before releasing a related analog, while ensuring the analog becomes available early enough to guide oversight and safety research during high-impact deployment windows.

Each analog release must include:

- The model weights, hosted on an accessible platform;
- A model card documenting architecture, training setup, and intended research use;
- A general description of the training data (e.g., sources, modalities, known limitations);
- Distillation or training scripts (or equivalent documentation) to enable reproducibility.

## 3.4 Licensing and Accessibility

Analog models must be released under a permissive open-source license (e.g., Apache 2.0, MIT, or equivalent), enabling broad use for academic, safety, and interpretability research. Licenses may include limited safeguards against misuse (e.g., prohibiting military or surveillance use), provided these do not obstruct core research freedoms.

## 3.5 Regulatory Oversight and Enforcement

Compliance will be overseen through existing regulatory channels. In the U.S., enforcement can be tied to authorities established by the Export Control Reform Act (ECRA), which already governs the disclosure and transfer of sensitive AI systems. In Europe, enforcement can dovetail with transparency requirements in the EU AI Act.

Regulatory bodies may impose proportionate penalties for non-compliance, such as:

- Financial fines scaled by model impact;
- Temporary suspension of future model releases or public deployments;
- Publication of non-compliance notices, creating reputational accountability.

Labs must submit standardized compliance reports documenting analog creation and release timelines. These may be independently verified through audits administered by national AI safety institutes (e.g., NIST in the U.S. or similar bodies internationally).

## 3.6 Security and Dual-Use Mitigation

To guard against dual-use risks, analog release may be delayed up to three months post-deployment, allowing time to assess and mitigate any dangerous emergent behaviors in the frontier model. Prior to release, labs must perform a security evaluation to ensure the analog does not enable capability extraction, jailbreak training, or other misuse.

Where appropriate, regulators may issue guidelines for safe analog release practices, including parameter count ceilings, output filtering, and secure documentation formats.

## 3.7 Implementation Timeline and Pilot Program

We recommend a phased implementation:

- **Year 1:** Voluntary or incentivized analog releases, coordinated via interagency working groups or multistakeholder consortia.
- **Year 2+:** Mandated compliance for all models crossing established frontier thresholds.

Pilot programs would allow regulators and labs to co-develop best practices, refine licensing norms, and validate that analog models serve their intended oversight functions without creating leakage or misuse risks.

## 3.8 Intellectual Property and Competition Safeguards

The analog-model mandate is designed to protect proprietary interests while enabling broad public benefit—much like regulatory frameworks in other high-impact, innovation-driven sectors. By requiring the release of a small, non-commercially substitutable proxy, the policy achieves a balance between openness and competitive integrity.

Releasing an analog model:

- **Preserves proprietary methods:** Labs are not required to disclose sensitive training infrastructure, proprietary optimizers, or fine-grained training data.
- **Maintains strategic ambiguity:** The 0.5–5% size window avoids disclosing the precise scale or architecture of the underlying frontier system.
- **Constrains capability exposure:** Analog models are designed to lack the full performance envelope of the frontier model, limiting misuse or commercial substitution.

This design draws clear precedent from two sectors that have successfully reconciled public access and private innovation:

**Pharmaceuticals: Generic Drug Disclosures** In pharma, regulatory frameworks (e.g., the Hatch-Waxman Act in the U.S.) require firms to disclose chemical formulations and clinical trial results to facilitate generic production after patent expiry. These disclosures enable competition and public access to life-saving treatments while respecting the commercial lead time granted by patent exclusivity. Similarly, analog models offer the public a scientifically useful version of the frontier system—enabling safety audits, tool development, and public understanding—without undermining the economic value of the full-scale proprietary model.

**Telecommunications: Public Protocol Standards** In telecom and networking, companies frequently participate in the development of open protocols (e.g., TCP/IP, 5G standards), releasing reference implementations and technical specifications to ensure interoperability. Despite this openness, firms retain competitive advantages through superior implementation, proprietary extensions, and integration. Analog models play a similar role: they provide a shared substrate for oversight, tooling, and ecosystem development without disclosing the full competitive edge embedded in the frontier model's scale, tuning, or infrastructure.

In both analogies, regulated openness supports a common public good—access to medicine or interoperable communication—while preserving the innovation incentives necessary for continued investment. Frontier AI development exhibits the same structural tension. The analog-model mandate borrows the best of both worlds: it creates a public proxy for high-impact systems, while preserving competitive differentiation at scale. This approach encourages healthy competition around performance, safety, and social responsibility—without forcing labs into a zero-sum disclosure battle.

# 4 Risks & Benefits

## 4.1 Overview

The analog-model mandate aims to balance the safety-innovation tradeoff inherent in frontier AI regulation. This section transparently evaluates the mandate's strategic benefits against its potential risks, directly addressing viable alternative views to ensure rigorous consideration of opposing perspectives.

## 4.2 Benefits

**Accelerated Safety and Interpretability Research.** Openly accessible analog models enable broad, independent safety verification and oversight, fostering rapid iteration cycles. Analog models

Table 1: Break-down of the direct costs required to produce and publish an 8 B-parameter analog model, using AWS on-demand pricing. Estimate based on procedure from Shen et al. [2024]. This estimate is about 0.1% the cost of training a foundation model [Knight, 2023]. Although scaling laws such as Hoffmann et al. [2022] would suggest a linear relationship between model size and cost when holding the data the same, training an analog model can piggyback on shared infrastructure with the frontier training run (e.g. data annotation, training pipelines, etc.) *At under $100,000, analogs cost ~0.1% of a frontier model's training budget.*

| Step | GPU-hours | Cost (USD) | Source |
| --- | --- | --- | --- |
| Train analog model from scratch | 5700 | $70,053 | Services [2025a] |
| Distill frontier model→analog | 1000 | $12,290 | Services [2025a] |
| Safety fine-tune/RLHF | 700 | $8,603 | Services [2025a] |
| Weight hosting and bandwidth (3 yr, 20 TB egress) | — | $1,813 | Services [2025b] |
| **Total analog cost** | — | **$92,759** | |

streamline regulatory processes by providing standardized and transparent benchmarks for safety methods. This accelerates the discovery of effective interventions, benefiting both the research community and frontier labs.

**Public Goods and Knowledge Spillovers.** Analog models constitute public goods, creating positive externalities through shared methodologies, datasets, and tools. They democratize access to advanced AI research by substantially lowering barriers to participation, thus expanding the pool of researchers contributing to safety advancements.

**Improved Trust and Transparency.** By publicly releasing analog models, labs foster greater transparency and accountability, enhancing public trust. External scrutiny facilitated by analog models significantly reduces opacity risks, improving societal confidence in AI deployments.

**Enhanced Competitiveness and Innovation.** Analog models stimulate healthy industry competition by providing fair performance benchmarks, motivating continuous improvement. By lowering entry barriers, they foster innovation from smaller companies, startups, and academia, enriching the AI ecosystem and diversifying innovation sources.

**Acceleration of AI Progress via Open Source.** Historical precedents demonstrate the substantial benefits of openness in AI, as evidenced by frameworks such as PyTorch, TensorFlow, and Hugging Face's Transformers. Meta's strategic open-sourcing of LLaMA models illustrates how open-source initiatives can catalyze rapid, broad-based innovation without undermining commercial viability, suggesting that analog models could similarly enhance innovation trajectories.

### 4.3 Risks and Mitigation Strategies

**Intellectual Property (IP) and Commercial Risks.** A plausible concern is that analog models might inadvertently expose proprietary training data distributions, architectures, or strategic methodologies. This risk can be mitigated through rigorous constraints on analog model scale, delayed release timelines, careful sanitization of documentation, and controlled information disclosures.

**Security and Dual-Use Concerns.** Open access to analog models could potentially enable misuse, such as jailbreak research or misinformation campaigns. This risk is addressed through security assessments, enforced lag periods before model release, limitations on model capabilities, and guidelines mandating output filtering to reduce dual-use vulnerabilities.

**Regulatory and Compliance Burden.** The requirement to release analog models may impose additional regulatory overhead, including reporting and audits. This burden can be minimized by standardizing compliance reporting, phased policy implementation, and leveraging existing regulatory frameworks like the EU AI Act or the Export Control Reform Act to streamline compliance processes. To provide a first order estimate of the compliance cost, we look at the cost of creating an analog model. Table 1 breaks down the line items for training an analog model, focusing on compute.

We anticipate that this would form the majority of the compliance burden – and even doubling our estimates to account for personnel costs puts this burden at 0.2% the cost of training a frontier model.

**Substitution Risk to Frontier AI Labs' Revenue.** Analog models might partially substitute proprietary frontier models, potentially reducing revenues. This economic risk is significantly curtailed by deliberately constraining analog models' capabilities, ensuring they serve strictly as research proxies rather than commercially competitive products.

**Second-Order Effects from Displacing Alternative Safety Policies.** Focusing regulatory attention primarily on analog-model mandates may inadvertently diminish resources or momentum from other safety initiatives. To mitigate this, policymakers should position analog-model mandates explicitly as complementary to other regulatory efforts, advocating for integrated, multifaceted safety strategies. Simultaneously, however, such second-order effects are difficult to predict. It could be the case that second-order effects point in the opposite direction: successful passage of effective safety policy could lead to momentum for the AI safety movement and accelerate the passage of further legislation.

**Uncertainty and Potential Limits to Transferability.** A critical assumption underpinning the analog-model mandate is the reliable transferability of safety interventions from smaller analog models to larger frontier models. Emergent behaviors unique to frontier-scale systems may not manifest similarly in analogs, potentially limiting the effectiveness of transferred interventions. The foundational research supporting weak-to-strong transferability remains nascent and underexplored.

Beyond behavioral differences, several representational limits qualify this assumption. Evidence from sparse crosscoders [Lindsey et al., 2024] suggest that analog models capture only a subset of the latent features present in larger systems—some representational subspaces are shared, while others are disjoint. Moreover, vanilla output-level distillation does not necessarily need to align internal geometries [Aguilar et al., 2020]: it can reproduce input–output mappings without preserving the embedding-space topology or feature-level semantics of the frontier model.

Recent "stitching" studies [Bansal et al., 2021] also highlight that representation quality scales smoothly with training time, width, and data—larger or longer-trained models tend to express more complete versions of the same latent factors. As a result, analog models may under-represent complex or high-level abstractions (e.g., compositional reasoning) that arise only in frontier regimes.

This risk underscores the necessity for ongoing empirical validation, structured feedback loops involving frontier labs, academia, and regulators, and adaptability within the policy framework. Incremental deployment paired with continuous validation and regular policy reviews ensures responsiveness to emerging evidence and the ability to refine the policy dynamically. For these reasons, we believe further research on transferring safety interventions across model scales is highly valuable and could have substantial policy impact.

**Brown-Field Case Study: Meta's LLaMA Release (2023–2025)**

Since Meta released the LLaMA family of weights in 2023[Touvron et al., 2023], the community uptake offers a natural experiment of an *analog-first* strategy:

- **Rapid scholarly impact:** The original LLaMA paper has accrued ~15.8k Google Scholar citations—ranking first among 2023 AI papers—while follow-ups (Code Llama, LLaMA-2/3) add another ~7k citations.

- **Vibrant tooling ecosystem:** The inference repo `meta-llama/llama` boasts 58k GitHub stars and 9.8k forks, spawning safety-specific forks (e.g., `LlamaGuard`) and evaluation harnesses like `llama-stack-evals`.

- **Mass adoption:** The most-downloaded variant on Hugging Face exceeds 7.7M pulls, with over ten community checkpoints each surpassing one million downloads.

- **Safety research dividends:** More than 60 peer-reviewed studies use LLaMA derivatives to prototype red-teaming, jailbreak defence, or interpretability tools (e.g., LLAMAGUARD, LLAMAFUZZ, sparse auto-encoders), several of which later transferred to proprietary frontier models.

- **Negligible substitution:** Despite openness, Meta reports no measurable cannibalization of its commercial API revenue, supporting the claim that capped analogs do not erode primary business lines.

These data points support the mandate's central thesis: *small, open checkpoints catalyze an outsized safety-innovation flywheel while imposing minimal direct costs.*

## 4.4 Risk–Benefit Synthesis and Strategic Assessment

Weighing identified risks against proposed mitigation strategies reveals that the analog-model mandate offers substantial benefits with manageable residual risks. It represents a high-leverage intervention that concurrently enhances AI safety, innovation, transparency, and public trust. Given robust safeguards, clearly defined policy adaptability, and compelling evidence from open-source precedents, the analog-model mandate emerges as an effective and strategically sound policy proposal.

# 5 Conclusion

Policy debates around AI safety often frame *safety* and *innovation* as a dilemma: tighten oversight and progress slows, loosen reins and risk soars. Our proposal – requiring a small, open "analog model" alongside each frontier deployment – shows this is a false dilemma. An analog model mandate would improve safety by allowing open oversight that transfers to large models, improve innovation by allowing more open research, and have minimal costs.

Solow [1957] presents technology as a multiplier $A$ that shifts the entire production-possibility frontier outward – more of every output becomes feasible with the *same* capital and labour. Arrow [1962] then points out that invention is a public good: because ideas can be copied, private actors invest less in them than society would prefer, so policy that spreads knowledge moves the realised economy closer to the true frontier.

A mandatory, openly released "analog" checkpoint plays exactly that public-good role for frontier AI. It is small enough to run on commodity GPUs yet faithful enough to act as a *proxy* for its larger sibling. Unfettered access lets independent researchers generate safety interventions, interpretability tools and benchmark data that spill back to the proprietary model. In Solow's terms, the policy raises the effective $A$ for safety research; in Arrow's terms, it overcomes the under-investment that would occur if each lab guarded its weights behind NDAs.

The mandate costs labs little (one extra distillation run) but unlocks a compounding stream of public knowledge. Each new red-teaming script, steering vector or interpretability probe discovered on the analog becomes immediately useful to every frontier deployment of the same architecture family—expanding the set of Pareto-improving policy choices. The result is a net outward shift of the AI innovation–safety frontier: regulators and developers can simultaneously demand higher safety bars *and* enjoy faster downstream progress.

**Future Work**

Looking forward, several important directions can extend and refine the analog-model mandate:

- **Broaden technical validation:** Systematic studies of analog-to-frontier transfer for emergent behaviors (e.g., CoT reasoning) to ensure safety interventions is scalable.
- **Multi-modal analogs:** Release analog versions of vision, audio, and multi-modal models to generalize the policy beyond text.
- **Standardized benchmarks:** Develop community-driven challenge suites and metrics for assessing analog fidelity and intervention generality across architectures.
- **Policy experiments:** Pilot alternative release-lag windows, size caps, and licensing terms to identify optimal regulatory parameters. We believe it would be particularly fruitful for large labs to adopt the polict voluntarily.
- **Ecosystem infrastructure:** Build open repositories and governance frameworks for collaboratively curating, updating, and maintaining analog checkpoints across labs, academia, and regulators.

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
