# OpenReview forum: "Position: Require Frontier AI Labs To Release Small "Analog" Models"
_NeurIPS.cc/2025/Position_Paper_Track — NeurIPS 2025 Position Paper Track_

### Official Review · Reviewer_266j · 2025-08-12

**Significance:** 4
**Presentation:** 4
**Rating:** 4
**Confidence:** 4

**Summary:**

This work argues organizations that develop "frontier" AI models ought to release smaller distilled "analogue" models with very similar architecture, trained on the same dataset and optimization objectives but with only a small fraction of the number of parameters. Such analogue models would supposedly facilitate interpretability research, since distilled models can likely still enable intervention studies given certain observed universality in LLM circuit. Additional benefits include promoting open source science and encouraging AI innovation which would bring societal benefits. At the same time, there are also risks associated with requiring to release analogue models, such as intellectual property loss and

**Strengths:**

- The work is easy to follow. Positions and counterpoints are clear presented and well argued for.
- The topic is very important in that it could potentially accelerate generative AI safety research.

**Weaknesses:**

- The main premise of the paper is that distilled models has "reliable transferability of insights" (L 46), i.e. (enough of) the "safety and interpretability interventions discovered" (L 61) in the analogue can be transferred to the base/teacher model which is commercially released. But this ignores [recent](https://arxiv.org/abs/2505.11837) [examples](https://aclanthology.org/2025.acl-short.61/) of distilled models having distinct behaviours (e.g. [amplifying harms](https://arxiv.org/pdf/2505.24842)) than the base model: for example, the distilled model may only capture part of the mechanisms of the base model, and so interventions found on it should not be expected to necessarily work on the base model. *In sum, the distilled -> base model transfer is not straightforward and claims need to be assessed empirically.*
- The analogy drawn on generic vs brand-name drug is not entirely accurate. In that case, the differences lie primarily in formulation, including inactive ingredients with potential side effects. For drugs needing precise dosage to function properly, this could make a large difference.

**Questions:**

- How would the proposed approach apply to closed-source models such as ChatGPT?
- Are there other ways to "democratize access to advanced AI research" (L256-257)? For example, grant better cloud GPU/TPU access to researchers who would otherwise have difficulty getting access to those machines?
- The discussion has mostly focussed on LLMs. Would it generalize to other kinds of generative AI models especially multimodal models?

**Alternative Position:**

Yes, and alternative positions are well-considered and addressed by the argument

**Author Identification:**

No.

**Context:**

2

**Discussion:**

3

**Ethics:**

["NO or VERY MINOR ethics concerns only"]

**Position:**

Yes, the paper argues for or against a position related to machine learning.

**Support:**

2

**Thoroughness:**

5

---

### Official Review · Reviewer_Gw3u · 2025-08-13

**Significance:** 3
**Presentation:** 3
**Rating:** 6
**Confidence:** 3

**Summary:**

This position paper proposes a regulatory mandate that any lab releasing a frontier AI model must also release a small-scale, openly accessible “analog model” trained on the same data and objectives, typically via distillation. The analog would be capped at 0.5–5% of the parameter count of the frontier model, released within 1–3 months of deployment, and licensed permissively for research. The authors argue that safety interventions and interpretability findings discovered in these smaller models reliably transfer to the larger systems, citing empirical studies, representational similarity research, and scaling laws. They present detailed policy mechanisms, compliance timelines, enforcement pathways, and cost estimates showing minimal burden. The paper also anticipates and addresses objections (IP, dual-use, substitution risk), drawing analogies from pharmaceuticals and telecom standards to illustrate that regulated openness can coexist with innovation incentives. The core claim is that such a mandate would relax the perceived safety–innovation tradeoff, providing substantial public benefit at low cost.

**Strengths:**

Presents a clear, concrete, and implementable policy proposal, backed by both technical research and policy precedent.

Effectively synthesizes empirical results on cross-scale transferability, representational similarity, and scaling laws to support the central claim.

Anticipates major objections (IP, competitive risk, security) and offers detailed mitigation strategies.

Uses compelling analogies from other regulated sectors to illustrate feasibility and precedent for openness without undermining innovation incentives.

Includes a cost analysis that convincingly argues for minimal compliance burden, enhancing the proposal’s practical appeal.

**Weaknesses:**

The discussion of cross-scale transfer largely assumes that smaller analogs capture the key behaviors of frontier models, without deeply engaging with cases where emergent or scale-specific behaviors may not manifest in the smaller version.

The empirical foundation for reliable safety transfer is still relatively early-stage; potential policy downsides if this assumption fails could be explored more fully.

The paper could give more attention to challenges of enforcing such a mandate globally and ensuring consistent compliance across jurisdictions.

**Questions:**

How would you ensure that analog models are representative enough when certain capabilities or behaviors only emerge at larger scales? Some concrete discussion of this is necessary for the arguments to land well.

Would you propose formal benchmarks or fidelity metrics to verify that analog models are suitable for meaningful safety and interpretability research?

What adaptations would be necessary for applying this mandate to multi-modal or non-text-based frontier models, where scaling patterns and representational alignment may differ?

**Alternative Position:**

Yes, and alternative positions are well-considered and addressed by the argument

**Author Identification:**

No.

**Context:**

4

**Discussion:**

3

**Ethics:**

["NO or VERY MINOR ethics concerns only"]

**Position:**

Yes, the paper argues for or against a position related to machine learning.

**Support:**

4

**Thoroughness:**

3

---

### Official Review · Reviewer_ActV · 2025-08-13

**Significance:** 3
**Presentation:** 3
**Rating:** 6
**Confidence:** 3

**Summary:**

The authors propose a regulatory policy requiring AI labs to release an “analog model”—defined as a small, openly accessible proxy for a proprietary model crossing established frontier thresholds—shortly after each major model deployment. Drawing on empirical evidence, the paper argues that safety and interpretability interventions developed on these smaller models reliably transfer to larger ones, as capability and representations scale predictably with model size in identical or closely matching architecture families. The authors conclude that their analog-model mandate could shift the safety–innovation frontier outward, reconciling public oversight with rapid private-sector AI progress.

**Strengths:**

This position paper is a thoughtful and timely contribution to AI safety research. The authors convincingly argue that the benefits associated with their proposed policy include accelerated safety research, improved transparency and public trust, and wider innovation participation at low cost (≈0.1–0.2% of training expenses), while risks such as intellectual property leakage or dual-use/misuse are addressed through size caps, delayed releases, and security assessments.

**Weaknesses:**

The potential limits to the transferability of safety and interpretability mechanisms from smaller analog models to larger frontier models could be explored more rigorously.

**Questions:**

Do capability and safety scale predictably for vision/audio/multi-modal models?

**Alternative Position:**

Yes, and alternative positions are well-considered and addressed by the argument

**Author Identification:**

No.

**Context:**

3

**Discussion:**

4

**Ethics:**

["NO or VERY MINOR ethics concerns only"]

**Position:**

Yes, the paper argues for or against a position related to machine learning.

**Support:**

3

**Thoroughness:**

3

---

### Note · Authors · 2025-08-28

**1-10 Additional Comments:**

None

**1-11 Submit Again:**

Probably yes

**1-1 Submission Process:**

3

**1-2 Next Year:**

No comment

**1-3 Future Development:**

The ability for authors to engage with reviewers on their reviews would be beneficial.

**1-4 Interest:**

["Panel discussions with other position paper authors", "Mentorship programs for early-career researchers"]

**1-4 Other Interest:**

No comments

**1-5 Thoughtful:**

9

**1-6 Supportive:**

8

**1-7 Technical Aspects Versus Position:**

5

**1-8 Gate Keeping:**

8

**1-9 Camera Ready Changes:**

Absolutely. The useful feedback from the reviewers will result in multiple changes to the position paper.

**3-1 Review Response1:**

ActV

**3-2 Reaction To Review1:**

We thank the reviewer for their thoughtful feedback.

**RE Do capability and safety scale predictably for vision/audio/multi-modal models?**
Safety and interpretability in multi-modal models remain less developed than in LMs. Early findings suggest universality across modalities [1], with preliminary evidence that steering vectors from distilled text-to-image models transfer to parent models [2]. While this doesn't establish reliable cross-modal transfer for all safety interventions, it indicates promising research directions.

This uncertainty supports gradual, research-informed implementation rather than immediate universal requirements. Initial policies could focus on text-dominant multi-modal models, expanding as understanding deepens.

The principle remains: ensure meaningful research access, establish safety boundaries, and iteratively build evidence on cross-scale/modal transfer.

[1] *The Platonic representation hypothesis* https://arxiv.org/abs/2408.07057
[2] *CASteer: Steering diffusion models* https://arxiv.org/abs/2503.09630

**RE Transferability limits need more rigorous exploration:**
We agree transferability cannot be assumed universally. We'll add a section addressing limits by:
* Surveying empirical evidence of transfer successes/failures (representational similarity studies, steering vector transfer, but also divergence cases in distilled models [3])
* Differentiating transfer levels: representation similarity, safety mechanism alignment, and scale-emergent behaviors
* Proposing fidelity benchmarks ensuring analogs are sufficiently representative
* Framing the mandate iteratively: analog release as ongoing research where limits are tested and understanding improves

[3] *Towards understanding distilled reasoning models* https://doi.org/10.48550/arXiv.2503.03730

**3-3 Review Response2:**

Gw3u

**3-4 Reaction To Review2:**

We thank the reviewer for their thoughtful feedback.

**Re: How would you ensure analog models are representative when capabilities emerge at larger scales?**
Our approach acknowledges analog models cannot perfectly replicate all large-scale behaviors. However, we propose a structured validation framework:

First, gradual adoption by major labs would create empirical foundations for understanding transfer limitations. Systematic documentation of which insights transfer successfully would build knowledge guiding researchers toward transferable directions.

Second, we recommend developing analog models at multiple intermediate scales—a "scaling ladder" approach helping identify capability emergence points and providing better coverage of the capability spectrum.

Third, even partial transferability represents significant progress over current situations where safety research often lacks model access entirely. Analog models complement, not replace, frontier model research.

**RE: Formal benchmarks or fidelity metrics for analog model suitability?**
Yes, we strongly support developing formal benchmarks. We propose multi-dimensional evaluation:

*Behavioral fidelity metrics* assess how well analogs reproduce key behaviors (deception detection, alignment failures, emergent reasoning). Include task-specific benchmarks and behavioral similarity scores.

*Mechanistic fidelity metrics* evaluate interpretability finding transfer between models, comparing activation patterns, attention mechanisms, or circuit analyses across scales.

*Research utility metrics* directly measure whether safety techniques developed on analogs improve frontier model performance, creating direct feedback loops for validation.

Establish benchmarks through collaboration between developers and AI safety community to capture critical research needs.

**RE: Adaptations for multi-modal/non-text frontier models?**
Please refer to our response to reviewer ActV.

**3-5 Review Response3:**

266j

**3-6 Reaction To Review3:**

We thank the reviewer for their well thought out feedback.

**Re How would the proposed approach apply to closed-source models such as ChatGPT?**
We propose requiring closed-source providers to release open-source analog models that meet established fidelity benchmarks, ensuring they enable meaningful safety research rather than serving as token compliance models.

Independent researchers would use these analog models to identify safety issues through multiple pathways: direct reproduction on frontier models for issues accessible via APIs (such as prompt injection attacks or jailbreaks), and collaborative disclosure to providers for issues requiring internal access (like training dynamics problems or subtle alignment failures).

To incentivize productive cooperation, companies would operate structured vulnerability reward programs similar to cybersecurity bug bounties. These would provide financial incentives for high-quality safety research, early disclosure periods allowing companies to address critical issues before public release, and clear protocols protecting researchers while building community knowledge through published findings.

Independent auditing mechanisms would verify that analog models maintain sufficient research utility and that vulnerability programs operate transparently with meaningful rewards. This framework transforms the current adversarial dynamic between safety researchers and companies into collaborative safety improvement, while ensuring public oversight and accountability through mandated analog model access.

This approach balances legitimate commercial interests with the critical societal need for safety research on the most capable AI systems.

**Re Would the approach generalize to other kinds of generative AI models especially multimodal models?**
Please refer to our response to reviewer ActV who asked the same question.

---

### Meta-Review · Area_Chair_FeUE · 2025-09-18

**Rating:** 6
**Confidence:** 4

**Strengths:**

Reviewers found the paper to be clear. It takes a coherent and novel stand, namely that makers of frontier AI models should also release scaled-down analog models for research purposes along with every new release. The paper carefully considers both technical and socio-technical aspects of the proposal, including calculation of the compliance burden, a nice touch.

**Weaknesses:**

Reviewers expressed concerns that emergent behavior might cause large models to behave differently from their small analog models. Although the authors address this concern, reviewers found it to be an area of continued question.

Reviewers also wondered whether the approach would carry over to multi-modal models.

**Questions:**

In responses to reviewers, the authors state, "Second, we recommend developing analog models at multiple intermediate scales—a "scaling ladder" approach helping identify capability emergence points and providing better coverage of the capability spectrum."

It would be great to hear more about how this would be implemented. Would this be a required part of every model release or only an occasional scientific benchmark check?

**Ethics:**

None raised

**Thoroughness:**

4

---

### Decision · Program_Chairs · 2025-09-26

Accept